# Co-isolation of genetically distinct *Burkholderia pseudomallei* strains from a single patient in North Queensland

Pauline M. L. Coulon[1]*, Piklu Roy Chowdhury[1], Ian Gassiep[2], Kay Ramsay[2], Aven Lee[3], Edita Ritmejeryte[3], Miranda E. Pitt[1], Joyce To[1], Sarah Reed[3], Patrick N. A. Harris[2], Garry S. A. Myers[1]

**1** Faculty of Science, Australian Institute for Microbiology and Infection, University of Technology Sydney, Ultimo, New South Wales, Australia, **2** Faculty of Health, The University of Queensland Centre for Clinical Research (UQCCR), Medicine and Behavioural Sciences, The University of Queensland, St Lucia, Queensland, Australia, **3** Mass Sperectometry Facility, The University of Queensland Centre for Clinical Research (UQCCR), Medicine and Behavioural Sciences, The University of Queensland, St Lucia, Queensland, Australia

* pauline.coulon@uts.edu.au

## Abstract

While investigating colony morphology variation within clinical isolates of *Burkholderia pseudomallei* (*Bp*) isolated from a single clinical infection, we identified two distinct strains (based on their multi-locus sequence type) from *Bp* TSV292. Reports of simultaneous co-infection are un-common, with only two studies in Thailand documenting such cases: (1) 2 out of 133 cases, and (2) 13 out of 59 patients infected with two to three polyclonal *Bp* strains exhibiting different genotypes. Here, we present the first comprehensive multi-omics characterization of co-infecting *Bp* strains isolated from a single infection at a hospital in Townsville, Australia, in 2018. This finding impacts the design of future diagnosis and treatment of *Bp*.

## Introduction

*Burkholderia pseudomallei* (*Bp*) is an opportunistic pathogen and the causative agent of melioidosis, a severe and often fatal disease in humans and animals [1,2]. Once considered endemic mainly to tropical and subtropical regions of Asia, Africa, Central and South America, and northern Australia [1], *Bp* is now recognized as established in the United States [3]. Diagnosis of melioidosis is challenging, as clinical presentations vary across geographical regions and laboratory misidentification can occur, especially when culture-dependent methods rely heavily on characterization of colony morphology on blood agar [4]. While *Bp* acquisition and infection will not cause melioidosis in healthy populations, people living with underlying conditions, such as diabetes, chronic kidney or lung disease, alcoholism, or compromised immune systems are at greater risk. In these populations, melioidosis often manifests as skin abscesses or severe pneumonia, which can progress to fatal sepsis with mortality

**Data availability statement:** Genomes from this study are available in GenBank under the Bioproject: PRJNA1295108. Proteomic raw data deposited to the ProteomeXchange Consortium via the PRIDE partner repository with the dataset identifier PXD066041.

**Funding:** This work was supported by the School of Life Sciences, University of Technology Sydney seed and AIMI seed fundings held by Dr Pauline M.L. Coulon. The School of Life Sciences, University of Technology Sydney strategic research accelerator funding (SRA 2726229) held by Prof Garry Myers. An EL2 Investigator grant from the National Health and Medical Research Council (APP2033851) held by Dr Patrick N.A. Harris. Royal Australasian College of Physicians Queensland Regional Committee Research Development Grant hold by Ian Gassiep allowed the collection and storage of *Bp* isolates. Experimental work was done at UQCCR, which would not have been possible without the International Society for Microbial Ecology sponsorship funding awarded to Dr Pauline M.L. Coulon.

**Competing interests:** The authors have declared that no competing interests exist.

rates reaching up to 52% [5]. With climate change intensifying extreme weather events such as heavy rainfall and flooding, the incidence of *Bp* infections is projected to rise [6]. Indeed, Queensland (Australia), has recently reported its worst year on record for melioidosis, with over 249 confirmed infections and 36 deaths linked to dispersal of the pathogen during flooding events (Queensland Health). While direct transmission between animals and humans has not yet been confirmed, growing concerns have emerged regarding potential transmission from pets, livestock or tropical animals to humans, either through contact with infected wounds or via shared exposure to contaminated environments [2,7]. Compounding these risks, *Bp* is naturally highly resistant to several antibiotic classes, and while frontline drugs such as meropenem, ceftazidime, and trimethoprim/sulfamethoxazole is still uncommon [8], emerging reports of resistance are increasing [9–12]. These clinical and epidemiological challenges highlight the urgent need for accurate diagnostics that account for the pathogen's remarkable adaptability and diversity.

A key yet often overlooked feature of *Burkholderia* species is their ability to undergo colony morphotype variation (CMV), a reversible process by which bacteria switch between distinct colony phenotypes [13]. This phenomenon reflects rapid adaptation to environmental stressors and is often associated with changes in outer membrane proteins, antimicrobial production, and virulence factors [14–18]. Mechanisms underlying CMV often include mutations in global regulators, two-component systems, genome reduction and duplication, bacteriophage integration, and DNA methylation (recently reviewed by Coulon and colleagues [13]). *Bp* is known to exhibit up to seven different macrocolony morphologies which can be modulated under laboratory conditions [14,18,19]. Importantly, CMV is not limited to adaptation *in vitro* as it also occurs during infection [20]. Approximatively 10% of 450 *Bp* clinical isolates produce mixed mucoid and non-mucoid CMV populations on blood agar [20]. In a long-term murine infection model, small colony variants (SCVs) were observed [21,22]. During human infection, differences in O-antigen lipopolysaccharides (LPS; [OPS]) production explains the mucoid phenotype (as observed on blood agar) even in the absence of mutations or altered expression of the *wbiA* O-antigen acetylase [20]. Additionally, genes within the LPS biosynthesis cluster are upregulated during persistent infection in mice [21,22]. The upregulation of the σ-54 dependent regulator *yelR*, is involved in the emergence of two reversible yellow *Bp* variants, which display enhanced resistance to hypoxic stress and increased colonization and persistence in the murine stomach [23].

These findings highlight that CMV can complicate both laboratory and clinical recognition of *Bp*. Distinguishing between true morphology variant and genetically distinct strains is particularly challenging, as both can present as colony diversity during culture. While investigating CMV within a cohort of *Bp* isolates [24] collected from clinical melioidosis cases at Townsville University Hospital (Australia) [25], we identified a polyclonal infection initially mistaken for colony morphotype variants. The apparent rough and smooth 'variants' were in fact two genetically distinct strains isolated from the same infection site. Such polyclonal *Bp* infection is rarely reported, with an apparent occurrence of 1.5% in a cohort of 133 patients when testing ten

colonies from each samples [26]. However, Kaewrakmuk and colleagues reported that 22% of *Bp* clinical infections are polyclonal, involving two to three different sequence types (STs; [27]). Importantly, no prior study has conducted an integrated multi-omics analysis of the co-isolated *Bp* pathogen. Here, we present the first study combining genomic, proteomic, and phenotypic analyses of two co-infecting *Bp* strains from a melioidosis patient in Australia, which underscores the need to refine diagnostic awareness and methodologies to detect clinically relevant polyclonality.

## Materials and methods

### Ethical approval

This study received ethical approval from the Royal Brisbane and Women's Hospital Ethics Committee (LNR/2020/QRBW/65573), with site-specific authority obtained from the Townsville Hospital and Health Service and approval under the Queensland Public Health Act.

This is a retrospective ethic approval from 1st January 1997–31st December 2020, and data were accessed/collected from December 2020 to April 2021. Patient consent was waived by the HREC.

### Colony morphology screening

*Bp* TSV292 was, originally, isolated from a blood culture that tested positive for *Bp*. The blood culture was used to set up a purity plate, and the resulting growth was used to make a −80 °C glycerol stock. For this study, *Bp* TSV292 from the frozen glycerol stock was streaked onto LB agar plates supplemented with 0.01% Congo Red (CRLA) and incubated at 37 °C for two consecutive overnight periods. Subsequently, four colonies of each observed morphological variation were subjected to further investigation.

### Genomic analysis

**gDNA extraction.**  Using either gDNA spin or high molecular weight gDNA extraction kits (NEB), gDNA was extracted from multiple macrocolonies grown for three overnights at 37°C on CRLA. Short-read sequencing (Illumina) was conducted on all CMVs. Briefly, libraries were prepared using 1 ng of gDNA via the Nextera XT kit and sequenced using the NovaSeq XPlus platform (35). On select isolates, long-read sequencing (Oxford Nanopore Technologies) was implemented. Libraries were prepared using 400 ng gDNA with the SQK-NBD114.24 kit and sequenced using R10.4.1 flow cells on the GridION platform. Reads were base called using Dorado 7.6.7.

**Genome assembly.**  Nanopore and Illumina reads were filtered and trimmed using respectively nanofilt (v.2.8.0; [28]) and fastp (v0.24.1; [29]). Each set of Nanopore long reads were then assemble using autocycler tool (v0.4.0; [30]). The long-read consensus assemblies were polished with Illumina short read sequences using polypolish (v0.6.0; [31]) and then pypolca (v3.0.1; [32]). Finally, the obtained assembly was annotated using batka (v1.11; [33]). For the second CMV isolated from each isolate, Illumina short reads were assembled using unicycler short read assembler (v.0.5.1; [34]) generating contigs. Illumina sequencing coverage was ~67.5x and 68x for TSV292_1 rough and TSV292_2 smooth respectively. Nanopore sequencing coverage was ~568x and 2,941x for TSV292_1 rough and TSV292_2 smooth respectively.

**Genomic modification analyses.**  Each set of Illumina reads, belonging to the second CMV, were mapped on their respective first CMV assembly using snippy tools with haploid settings (v4.6.0; [35] available on Galaxy) to determine SNPs and indels, to confirm the results, the first CMV Illumina reads were also mapped onto their own assembly, resulting in showing no genomic variation.

**Genotyping and phylogeny analysis.**  Six thousand two hundred fifty-nine allelic profiles of 1,225 available *Bp* isolate genomes on pubMLST genomes database were retrieved and used to generate a distance phylogenetic tree with GrapeTree using allelic profile differences [36]. The resulting tree was generated using ggtree R package. PhyloSift

v1.0.1 was used to resolve the subclade phylogeny of the two clusters (identified from the wgMLST clustering analysis) by computing a maximum-likelihood tree based on the 37 prokaryotic marker genes of each genome (available on pubMLST) using FastTree2.2 [37]. FigTree v1.4.4 was used to draw the unrooted phylogenetic trees and FastANI was used to identify the pairwise average nucleotide identity with the respective ST identified in this analysis. BRIG-0.95 [38] was used to map pairwise BLASTn alignment of selected genomes [39].

**Prediction of antibiotic resistance genes.** AMR gene orthologs present in genomes was identified using the CARD database [40] and outputs were filtered to tabulate data of candidates which had 100% coverage of subjects with at least 40% similarity. Antibiotic Resistance Detection and Prediction (ARDaP) tool was used to determined antibiotic resistance profile of both isolates [41].

**Whole genome alignment.** Both MAUVE [42] and LASTZ [43,44] tools were used to aligned TSV292_1 (rough) and TSV292_2 (smooth) together, using the default settings. To correct aligned genomes with LASTZ tool, the first residue numbers for each chromosome were changed as followed, to set up the same origin. For TSV292_1 rough contig 2 residue 1,249,185 became 1 while TSV292_2 (smooth) contig 1 was reversed and residue 285,486 became 1; TSV292_2 (smooth) contig 2 residue 490,391 became 1.

**Determination of unique and duplicated genes.** The sequence of annotated genes from each genome were aligned to the other genome's nucleotide sequences using BLASTn with the following parameters: -evalue 10 -outfmt 6 -num_threads 3 -max_target_seqs 1 -max_hsps 1 [39]. Hits with >90% sequence identity were retained and cross-analyzed with the corresponding results obtained from comparisons against *Bp* K96243. Finally, genes located within genomic regions identified as deleted or inserted by LASTZ were filtered out.

## Methylome analysis

Pod5 files obtained from Nanopore sequencing were merged, using pod5 merge function and analyzed for DNA methylation motifs using the Dorado with both dna_r10.4.1_e8.2_400bps_hac@v4.3.0_6mA@v2, dna_r10.4.1_e8.2_400bps_hac@v4.3.0_5mC_5hmC@v1 models (https://github.com/nanoporetech/dorado). To identify the location and number of detected motifs, as well as the genes encoding DNA methyltransferases, the call_methylation (using default settings --min_strand_coverage 10, --methylation_confidence_threshold 0.66, --percent_methylation_cutoff 0.66, and --percent_cutoff_streme 0.9) and annotate_rm functions from MicrobeMod [45] were used. These tools were provided with the corresponding genome assembly ".fasta" and annotation ".gff3" files for each isolate.

## Proteomic analysis

**Bacterial growth condition.** Individual colonies (three biological replicates) were picked and stabbed into CRLA plates and incubated at 37°C for three nights. The three resulting microcolonies were used for proteomic analyses.

**Protein extraction and peptide preparation.** Bacterial cells in 1M Tris-HCL containing 1% SDS were sonicated in an ultrasonic bath YJ5120−1 (Labtex) for 3 minutes, followed shaking at 1,000 rpm at 95°C for 15 minutes on a Thermomixer Compact (Eppendorf). The mixture was allowed to cool down to room temperature and 30 µL was mixed with 270 µL of Buffer I (6 M guanidine chloride, 50 mM Tris pH 8.0, 10 mM DTT) and incubated at 30°C for 40 min. Following addition of 9 µL of 0.5 M iodoacetamide, the sample was incubated at 30°C for 50 min. The reduced and alkylated protein sample was transferred to a 10 kDa cut-off Amicon Ultra device (Merck) which was placed into a collection tube and centrifuged at 14,000 × g for 30 min. The flow-through from the column was removed and 300 µL of 100 mM ammonium bicarbonate was added to the column followed by centrifugation at 14,000 × g for 30 min. To digest proteins in the column, 130 µl of 100 mM ammonium bicarbonate containing 2 µg Trypsin was added. The column was incubated at 37°C for 24 h and then centrifuged in a new collection tube at 14,000 × g for 30 min. Fifty µL of water with 0.1% formic acid was then added to the spin column followed by centrifugation at 14,000 × g for 30 min to elute the remaining digested peptides. Enzymes and salts were removed using $C_{18}$ ZipTips (Merck) following the manufacturer's instructions and desalted peptides were

eluted with 80% acetonitrile/0.1% formic acid into microcentrifuge tubes. Digested peptide samples were dried down using a vacuum concentrator (Concentrator plus, Eppendorf) and reconstituted with 50 μL water with 0.1% formic acid prior to LC-MS/MS analysis.

**LC-MS/MS analysis.** The samples were chromatographically separated and resulting proteomics data was acquired using an Acquity M-class micro-LC system (Waters, USA) coupled with ZenoTOF 7600 LC-MS/MS system using the OptiFlow Turbo V ion source. (AB Sciex). Two μL of sample were loaded onto Micro TRAP C18 column (0.3 × 10 mm; Phenomenex) and washed for 10 min at 10 μL/min with 97% solvent A (water with 0.1% [v/v] formic acid) and 3% solvent B (99.9% [v/v] acetonitrile, 0.1% [v/v] formic acid). Liquid chromatography was performed at 5 μL/min using a 2.7 μm Peptide C18 160Å; (0.3 × 150 mm; Bioshell) column with the column oven temperature maintained at 40°C. The gradient started at 3% solvent B which was increased to 5% by 0.5 min, followed by an increase to 35% solvent B by 38.9 min and then another increase to 80% solvent B by 39 min. The column was washed with 80% solvent B for 3.9 min before equilibrating with 3% solvent B from 43 to 45 min. Data was acquired using Zeno SWATH DIA, using a 65 variable width SWATH windows. The TOF MS parameters were as follows: precursor mass range was 400–1500 m/z; declustering potential (DP) was set to 80 V and accumulation time was set at 0.25 s. The TOF MS/MS parameters were as follows: fragmentation mass range was set to 100–1500 m/z; with an accumulation time of 0.025s; fragmentation mode was set to CID with Zeno pulsing selected. The collision energy for the MS/MS acquisition was automatically adjusted according to the m/z and charge of the peptide.

**Data analyses.** Raw label free quantification (LFQ) LC-MS/MS data were converted to.mzML files, using MS convert, to allow the data search using Fragpipe v22 [46]. Protein identification was conducted against the respective annotated proteomes using the DIA_SpecLib_Quant workflow, which builds a spectral library with MSFragger-DIA and performs quantification with DIA-NN. The output "diann-output.pg_matrix.tsv" was filtered to remove contaminants and retain proteins with at least two non-missing values for at least one group. Data was normalized using the variance stabilization normalization [47]. Proteins with two or more missing values in one group were categorized as "detected/not detected." For proteins with a single missing value in one group, missing data were imputed using the mean of the other replicates in that group. Differential protein expression between groups was assessed using Welch's t-test, with Benjamini–Hochberg correction applied to control the false discovery rate (adjusted p < 0.05) [48,49] (S5-S9 Tables).

**Proteomic representation.** To assign homologous proteins and annotations between TSV292 rough, smooth and K96243 (*Bp* model), were assigned as described in the "determine the number of unique genes" method. Clustering of orthologous groups (COGs) for annotated proteins were determined using EggNOG tool v5 [50]. To represent the data in a volcano plot, detected/not detected data were imputed using the lower normalized Log2 value for intensity with an imputed adjusted pvalue randomly assigned between 0.01 and 0.00001. Enrichment analysis was done using a Fisher exact test based on the COGs, using Claude AI for the script.

## Phenotypic assays

**Production of virulence factors.** Individual colonies (four biological replicates) were picked and stabbed into LB agar plate containing 1.5% skimmed milk (assessing proteolytic activity; [51]), Cas assay plates made with 10X chelated LB [24]; 0.5 g/L yeast extract supplemented with 4g/L D-mannitol (YEM) and agar (assessing mucoidy; [52]) and Columbia horse blood agar plate (assessing hemolysis activity; [53]). Plates were incubated at 37°C for 3 days. Inhibition zones were measured and analysed statistically. Data normality was assessed using the Shapiro–Wilk test, followed by either a Wilcoxon rank-sum test or a t-test to compare the phenotypic results between TSV292_1 (rough) and TSV292_2 (smooth). The reference isolate Bp K96243 was not included in our tests as, phenotypes are strain specific.

**Antimicrobial sensitivity assay.** Individual colonies were resuspended in 0.9% NaCl solution to obtain a turbidity of 0.5–0.63 McFarland. Bacterial suspensions were then spread onto Muller Hinton II agar plates and incubated overnight at 37°C with 0.002–32 μg/mL Meropenem (MEM) ETEST (Biomérieux), 10 μg Ceftazidime (CAZ) and 25 μg

Sulfamethoxazole-Trimethoprim (TMP-SMX) Antimicrobial Susceptibility discs (Oxoid). For the discs assay, antimicrobial susceptibility was determined by following the recommendation from EUCAST(v15; https://www.eucast.org/fileadmin/src/media/PDFs/EUCAST_files/Breakpoint_tables/v_15.0_Breakpoint_Tables.pdf). For CMVs from the same isolate showing considerable visual changes in antibiotic susceptibility, three biological replicates were conducted on independent days, and the average of inhibition data were used to determine their susceptibility characteristic. Data normality was assessed using the Shapiro–Wilk test, followed by either a t-test to compare the phenotypic results between TSV292_1 (rough) and TSV292_2 (smooth). The reference isolate *Bp* K96243 was not included in our tests as, phenotypes are strain specific.

## Results

### Isolation of TSV292_1 rough and TSV292_2 smooth colony morphotypes

In *Burkholderia,* CMV typically occurs upon removal of environmental pressure or during aging [14,18]. Because selective media for *Burkholderia pseudomallei* (*Bp*) often contain antibiotics that can induce colony morphology changes (e.g., the wrinkled phenotype in *Bp*), the clinical isolate TSV292 was streaked directly onto LB agar supplemented with 0.1% congo red (CRLA) from a −80 °C glycerol stock to facilitate CMV distinction. On this medium, a mixed population was observed, consisting of flat colonies with a red center (type RI) and smoother, raised colonies (type SI) (**Fig 1A**). The rough and smooth phenotypes were confirmed by isolating single colonies and subculturing them on CRLA or Ashdown agar (**Fig 1B**). Both media yielded CMV types consistent with those previously reported for isolate TSV82 [24].

Importantly, this initial observation suggested classic CMV, highlighting the risk of misclassifying polyclonal infections as morphotypic variation during standard diagnostic workflows.

### Polyclonal strains masquerading as CMV in melioidosis infection

Genomic variation, master regulator expression or epigenetics are known drivers of CMV in *Burkholderia*; hence, we first investigated whether SNPs or insertion and deletions (indels) accounted for the distinct morphologies of TSV292. Illumina reads from TSV292 rough were mapped against the assembled genome of TSV292 smooth, using Snippy [35], which revealed 23,229 genomic modifications (S1 Table). This unexpectedly high level of variation suggested that the two morphotypes were polyclonal rather than colony morphotype variants of the same isolate. Multi-locus sequence typing (MLST) further supported this conclusion: the rough isolate was identified as ST-2319 (TSV292_1; PubMLST sample ID:7631; previously reported by Gassiep and colleagues [25]), whereas the smooth isolate represented a novel sequence type (TSV292_2; PubMLST sample ID:7519; ST-2323) (S2 Table).

These findings demonstrate that rough and smooth variant colonies can mask genetically distinct strains, emphasizing the need for genomic-level analyses to accurately identify co-infections in melioidosis. Because simultaneous infection with multiple distinct *Bp* clones from the same patient is considered rare [26,27], we further investigated the genomic and proteomic profiles of these isolates to gain a deeper understanding of their divergence.

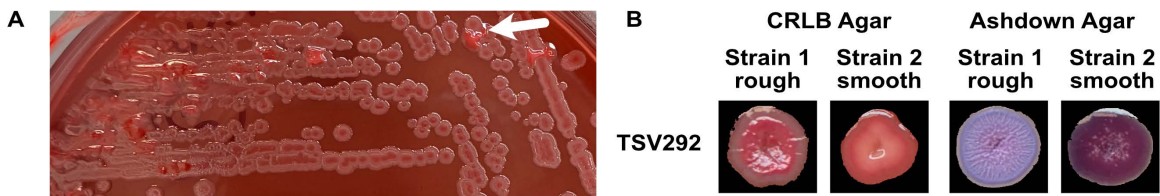

**Fig 1. Colony morphologies of the two *Bp* strains isolated from TSV292. A)** −80°C glycerol stock streak on CRLA, white arrow indicates the presence of second colony morphotype corresponding to TSV292_2 smooth strain. **B)** Colony morphology of both isolates after 3-days growth on both CRLA and Ashdown agar.

## Comparative analysis of *Bp* ST-2319 and ST-2323

The average nucleotide identity (ANI) between the two genomes was 99.4%. TSV292_1 (rough) had a genome size of 7,262,464 bp with a GC content of 68.1%, while TSV292_2 (smooth) had a genome size of 7,352,845 bp with a GC content of 68.0%. Clustering analysis of whole-genome MLST (wgMLST) based on 6,259 allelic profiles of 1,225 available isolate genomes in the PubMLST database (August 2025) revealed two distinctly separate groups (Fig 2A). Maximum-likelihood phylogenetic analysis showed that TSV292_1 rough clustered with *Bp* strains previously isolated from melioidosis patients admitted at Townsville University Hospital between 1997 and 2020 [25], whereas TSV292_2 smooth was more closely related to *Bp* strains isolated in Cambodia (Fig 2B). TSV292_1 rough and TSV292_2 smooth shared >99.2% ANI with their closely related *Bp* isolates but also contain variable regions representing strain-specific regions (S1 Fig).

Whole-genome alignment of the TSV292 co-isolates, using MAUVE algorithm [42], revealed synteny across the locally co-linear blocks and highlight genomic rearrangements likely as a result of recombination events in both chromosomes (S2 Fig). Alignment using the LASTZ tool [43,44], further revealed divergence between TSV292_1 rough and TSV292_2 smooth across several genomic regions, totalling 240,390 bp and 357,022 bp, respectively. In addition, 93 and 74 duplicated genes were identified outside these regions, which likely account for the observed difference in genome size (S3 and S4 Figs).

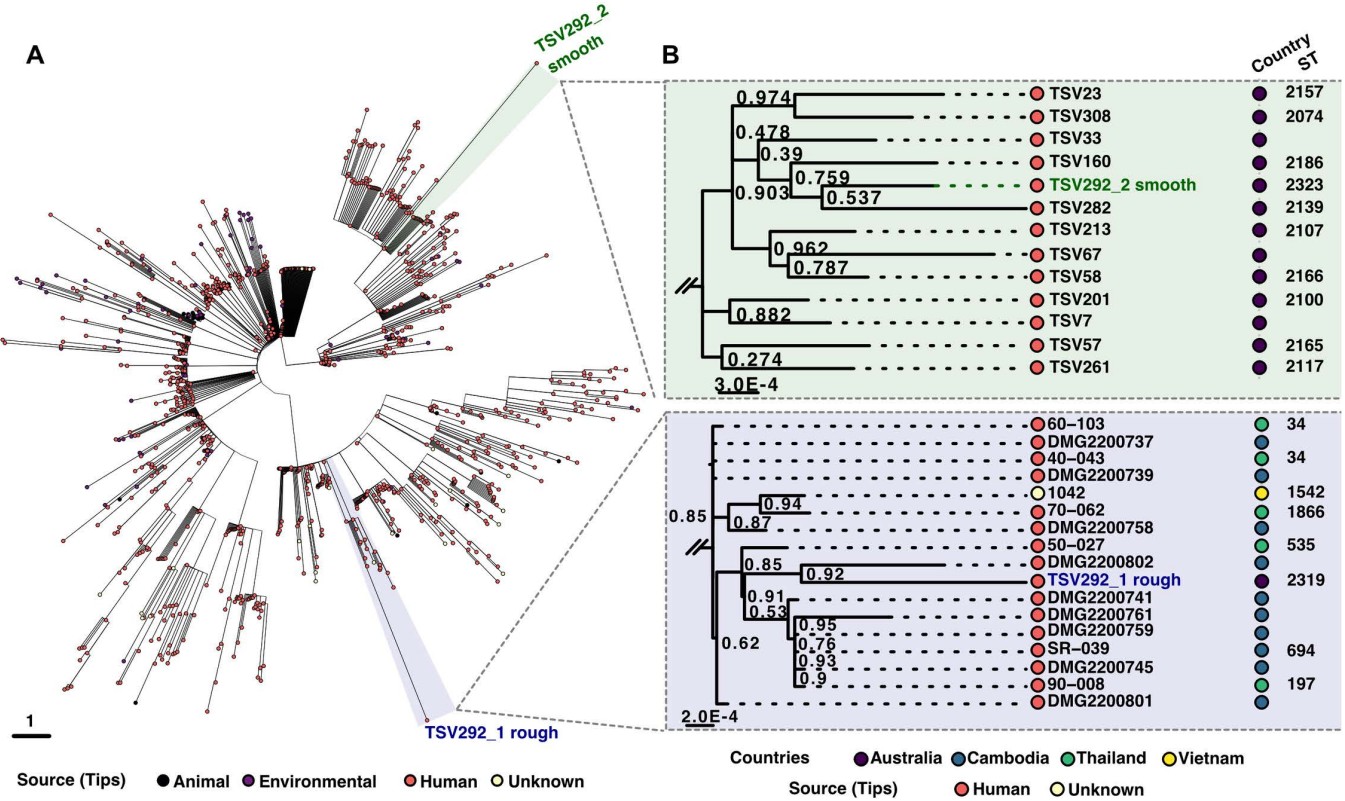

**Fig 2. Two distinct *Bp* strains co-infecting the same patient suffering from melioidosis in 2017. A)** Circular phylogenetic tree constructed using GrapeTree software showing relationships between isolates based on wgMLST allelic profiles. The tree represents a minimum spanning tree where branch lengths correspond to the number of allelic differences between isolates across genome loci. Each tip represents an individual isolate, with closer proximity indicating greater genetic similarity. Two reference strains are highlighted: TSV292_1 rough (blue) and TSV292_2 smooth (green). The scale bar represents 1 allelic difference. Isolates are colored by source as indicated in the legend. The analysis included 1,225 isolates characterized using 6,259 loci from the wgMLST scheme. **B)** Zoomed on closest isolated for both TSV292_1 rough (blue, right top panel) and TSV292_2 smooth (green, right bottom panel), using maximum likelihood phylogenetic analysis, using Fasttree, based on closest relative genomes.

In TSV292_1 rough, regions absent in TSV292_2 smooth included: (i) a 7,465 bp segment containing a type II restriction enzyme and DNA Methyltransferase (DNA MTase) predicted to be responsible for m5C methylation on GTCGAC pattern (TSV292_R_RS_00500–00501) surrounded by IS3 family transposase; (ii) a 36,612 bp region including predicted phage elements and a singleton DNA MTase (TSV292_R_RS_02319); and (iii) a 61,103 bp region with 26 putative transposases consistent with lateral gene transfer, that also encoded the *Bp* adhesion (*bpa*) gene cluster—implicated in bacterial attachment and immune evasion [54] —suggesting functional differences in host interaction and pathogenic potential.

On chromosome 2 TSV292_1 rough carried a 43,430 bp region with an integrase (TSV292_R_RS_04679) and phage elements; a 36,084 bp region rich in transposases and viral genes, including a type III restriction DNA methyltransferase (TSV292_R_RS_03252; S6 Table); a 21,283 bp region containing a putative origin of replication with a PAAR gene and a type III restriction DNA MTase (TSV292_R_RS_03525) suggestive of plasmid integration; and a 43,340 bp region containing phage elements (S3 and S4 Tables).

By contrast, TSV292_1 rough lacked multiple phage-associated and transposon-rich regions present in TSV292_2 smooth, including loci encoding type II, III, and IV restriction MTases predicted to modify m4C, m5C, and m6A motifs (e.g., TSV292_RS_03205, TSV292_RS_03345, TSV292_RS_15530, TSV292_RS_15535, TSV292_RS_18580, TSV292_RS_26185; S3-S6 Tables). Differences also extended to the flagellar biosynthesis cluster, where the two genomes displayed altered gene order (S5 Fig). As deletion of genomic regions commonly occurs during bacterial adaptation and host evolution [55], we next examined whether each genome carsried unique genes. To assess this, annotated genes from each genome were aligned against the other genome's nucleotide sequences using BLAST, retaining hits with >90% sequence identity, and cross analysed the results of both genomes blast against K96243. From this analysis, only one unique gene was identified in TSV292_1 rough (TSV292_R_RS_00840), encoding a hypothetical protein. These findings were consistent with the results of our LASTZ analysis (S3 and S4 Figs).

Taken together, these results indicate that divergence between both strains involved acquisition or loss of phage elements, transposons, plasmid-associated regions, and DNA methylation systems. Such changes not only underscore the genome plasticity of *Bp* but also suggest functional consequences for immune evasion, horizontal gene transfer, and regulation of surface structures critical for infection. Multi-omics approaches are therefore essential to resolve polyclonal infections and guide treatment strategies.

### Distinct phenotypes in antibiotic sensitivity, mucoidy, hemolysis, siderophore production and proteolytic activity

To confirm the proteomic analyses, phenotypic assays assessing virulence-related traits, including antibiotic susceptibility, were performed (**Table 1**) and revealed notable differences between the two isolates. TSV292_2 smooth exhibited a higher resistance to trimethoprim-sulfamethoxazole (TMP-SMX; 5.32 ± 1.15 mm against 23.88 ± 5.79 mm for TSV292_1 rough), with no apparent difference in predicted antibiotic resistance by ARDaP (S7 and S8 Tables; [41]), and displayed increased mucoidy, consistent with higher exopolysaccharide (EPS) production when grown on YEM agar. In contrast to TSV292_1 rough, the TSV292_2 smooth strain lost its ability to lyse red blood cells and lacked proteolytic activity. Interestingly, both isolates produced a smooth-edged sector around the main colony when grown on Ashdown and blood agar, suggesting an active role in CMV emergence (**Fig 3**). Differences in antibiotic sensitivity and production in virulence factors underscore that genomic analyses alone could mislead clinicians regarding virulence potential and drug susceptibility, highlighting the need to integrate molecular and phenotypic analyses in melioidosis diagnostics.

### Despite being two isolates, proteomics profiles characterise them as rough and smooth *Bp* CMV morphotypes

Proteomic profiling confirmed that the two isolates correspond to the expected profiles of rough and smooth *Bp* morphotypes, with differences in virulence- and resistance-associated pathways (Fig 4). In total, 1,015 proteins were differentially produced between the two isolates (Fig 4A; S9 and S10 Tables).

**Table 1. Phenotypic profile of both *Bp* TSV292_1 rough and TSV292_2 smooth.**

| Isolate | Colony morphology | Antibiotic Susceptibility test | | | Mucoidy | Hemolysis | Proteolytic Activity (cm²) | Siderophore production (cm²) |
|---|---|---|---|---|---|---|---|---|
| | | MEM | CAZ | TMP-SMX | | | | |
| TSV 292 | Strain 1 (rough) | S | I | I | + | beta | 3.26 ± 0.75** | 0.399 ± 0.093** |
| | Strain 2 (smooth) | S | I | R† | ++ | gamma | 0.88 ± 0.09 | 1.811 ± 0.33 |

EUCAST: S = Susceptible; I = Susceptible increased exposure; R = Resistant; Meropenem (MEM): S≤2; R>2; Ceftazidime (CAZ; disk 10 μg): S≥50mm; R<18mm; Trimethoprim-Sulphamethadoxone (TMP-SMX; disk 1.25–23.75 μg): S≥50mm; R<17mm;

"†" represent a TMP-SMX susceptibility statistically different between TSV292_1 and TSV292_2 using a ttest;

Mucoidy: + = mucoid; ++ = more mucoid;

Hemolysis at 8 days: beta = complete lysis of red blood cell; gamma = no lysis;

Proteolytic activity and Siderophore production: average ± stand deviation;

"**" represent statistical significance (ttest) in both proteolytic activity or siderophore production with a pvalue comprised between 0.05 and 0.01 when comparing both TSV292_1 and TSV292_2.

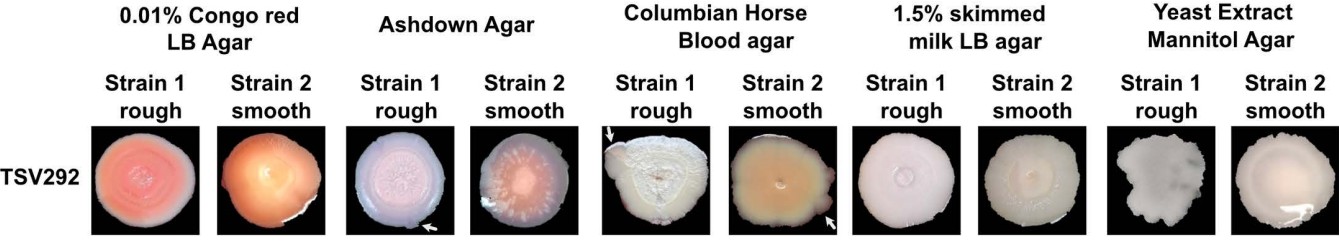

**Fig 3. CMV emergence from both isolates occur after 8 days of growth on various media.** White arrows indicate variants emergence from the main colony when culture on different media to study bacterial phenotypes.

In the TSV292_2 smooth strain, proteins associated with antimicrobial gene cluster biosynthesis were more abundant or exclusively detected. These included those involved in the production of pyochelin, malleobactin, malleilactone, bactobolin, and syrbactins [56–58]. Notably SyrC/GlbC, part of the syrbactin cluster, is also known to influence *Bp* survival and replication within immune cells [56,59]. Additionally, proteins related to flagellar biosynthesis, the quorum sensing system 2 (BpsR2; belonging to the bactobolin biosynthesis cluster), and the Hmq system, which produces HMAQs with antimicrobial activity and known interaction with QS systems, were more abundant or exclusively detected [58,60,61].

In contrast, the TSV292 rough morphotype showed higher abundance or exclusive detection of proteins involved in exopolysaccharide (EPS) production, type III O-polysaccharide (OPS) synthesis, DNA methyltransferase (MTase) responsible for the CACAG methylation motif, the quorum sensing system 3 (BpsI3/BpsR3), and the master regulator ScmR (**Fig 4B**).

Proteins from antibiotic resistance genes [12,41,62,63], predicted using the CARD database (S11 Table; [40]), were generally not detected or detected at lower abundance in TSV292_2 smooth compared with the rough isolate (Table 2). This includes both AmrR, the TetR family transcriptional regulator that controls the expression of the AmrAB-OprA efflux pump, and BpeT, the transcriptional regulator of the BpeEF-OprC efflux pump, potentially explaining the observed TMP-SMX resistance in the smooth strain in **Table 1** [41]. These results indicate a difference in adaptations between the two strains, with TSV292_2 smooth adopting an environmental-like smooth form and TSV292_1 rough displaying a pathogenic rough form [16,58]. Proteomic profiling further demonstrates that diagnosis based solely on colony morphology—or even coupled with genomic data—may fail to capture the full functional diversity of co-infecting strains. Detailed molecular characterization of protein expression can uncover hidden differences in virulence, antimicrobial production, and resistance mechanisms, providing a more reliable foundation for diagnostics and treatment decisions.

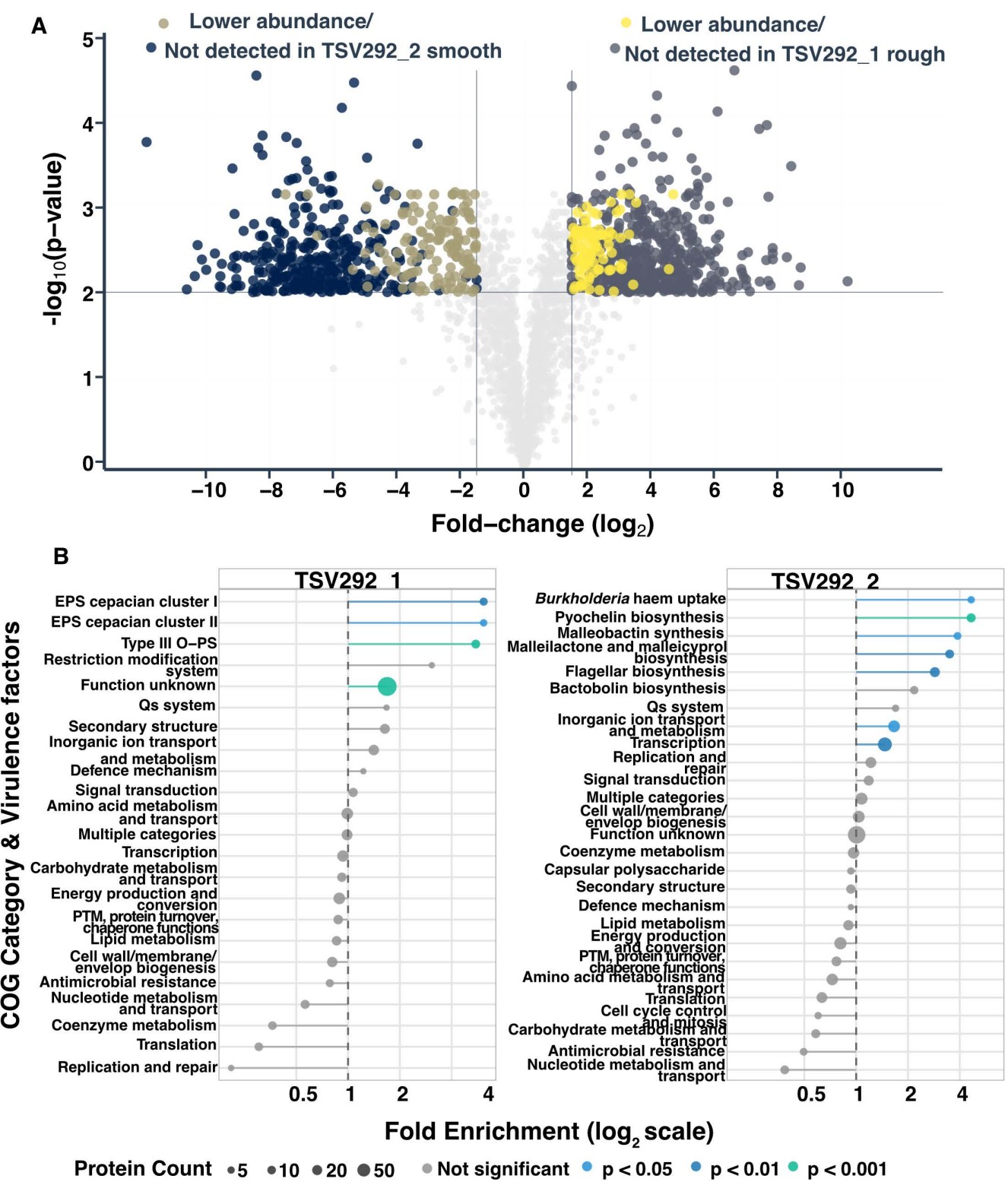

**Fig 4. Proteomic profiling of both TSV292_1 rough and TSV292_2 smooth isolates. A)** Volcano plot representing the significant abundance changes in proteins between both isolates. **B)** Enrichment analysis of virulence factors and COGs reflecting the proteins changes.

**Table 2. Significance differences in the abundance of protein involved in antibiotic resistance.**

| Protein in Bp K96243 | Protein in TSV292 R | Protein in TSV292 S | Gene Name | Product | Mean intensity in TSV292_1 rough (Log2) | Mean intensity in TSV292_2 smooth (Log2) | Differential abundance TSV292_1 rough vs TSV292_2 smooth (Log2) | pvalue (ttest) | Adjusted pvalue (fdr=0.01) | Significance | Resistance | Reference |
|---|---|---|---|---|---|---|---|---|---|---|---|---|
| BPSL1440 | RS_05656 | RS_24985 | | hypothetical protein | 14.4 | 12.2 | −2.178 | 0.007 | 0.019 | Lower abundance in rough | SXT | [62] |
| BPSL1357 | RS_04972 | RS_21485 | folP | dihydropteroate synthase | 11.3 | 13.0 | 1.729 | 0.001 | 0.006 | Lower abundance in rough | SXT | [63] |
| BPSS0946 | RS_02455 | RS_12455 | penA | beta-lactamase precursor | 12.5 | 14.4 | 1.903 | 0.001 | 0.005 | Lower abundance in rough | CAZ | [62] |
| BPSL0814 | RS_04343 | RS_18740 | bpeA | RND family acriflavine resistance protein A | 13.9 | 15.7 | 1.778 | 0.000 | 0.002 | Lower abundance in rough | MEM | [62] |
| BPSS1898 | RS_01406 | RS_02545 | fadH | 2,4-dienoyl-CoA reductase | NA | 9.1 | NA | NA | NA | Not detected in rough | MEM | [62] |
| BPSL1805 | RS_05301 | RS_23130 | amrR | TetR family regulatory protein | NA | 10.0 | NA | NA | NA | Not detected in rough | MEM | [12,62] |
| BPSS0290 | RS_00441 | RS_08815 | bpeT | LysR family transcriptional regulator | NA | 10.9 | NA | NA | NA | Not detected in rough | MEM/CAZ/SXT | [12,41,62] |
| BPSS1269 | RS_02119 | RS_00900 | syrC/glbC | peptide synthase/polyketide synthase | NA | 12.5 | NA | NA | NA | Not detected in rough | AMC/CAZ | [62] |
| BPSS1095 | RS_02298 | RS_13015 | | heat-shock chaperone protein | NA | 13.0 | NA | NA | NA | Not detected in rough | SXT | [62] |
| BPSS1654 | RS_01708 | RS_01070 | | cytochrome P450 | 9.4 | NA | NA | NA | NA | Not detected in smooth | AMC | [62] |
| BPSL0812 | RS_04342 | RS_18735 | bpeR | TetR family regulatory protein | 10.5 | NA | NA | NA | NA | Not detected in smooth | MEM | [12] |
| BPSS2246 | RS_01043 | RS_06515 | | sensor kinase/response regulator fusion protein | 11.2 | NA | NA | NA | NA | Not detected in smooth | SXT | [62] |
| BPSS1197 | RS_02196 | RS_00505 | | non-ribosomal peptide synthase | 12.3 | NA | NA | NA | NA | Not detected in smooth | CAZ/MEM/SXT | [62] |
| BPSL0816 | RS_04346 | RS_18755 | oprB | outer membrane efflux protein | 13.6 | NA | NA | NA | NA | Not detected in smooth | CAZ | [62] |
| BPSL2553 | RS_02674 | RS_28050 | | siderophore receptor protein | 10.5 | 13.4 | 2.850 | 0.007 | 0.018 | Lower abundance in smooth | CAZ/MEM | [62] |
| | RS_01304 | RS_05215 | blaOXA | OXA-42 family class D beta-lactamase | NA | | | | | | TET | |

*NA: not applicable.

## Methylome showed no overall difference in methylated sites between TSV292 rough and smooth

Predicted restriction enzyme methylated motifs were consistent with the one described by Nandi and colleagues [64] (S7 and S12 Tables). Given the observed difference in abundance of the DNA MTase responsible for the CACAG methylation motif (TSV292_R_RS_03660/03661 in rough colony morphotype TSV292_RS_15080/15085 in the smooth, for which proteins were not detected in TSV292_2 smooth while detected in TSV292_1 rough isolates), we further investigated the methylome profiles of both isolates (S12 Table). Despite the presence of distinct abundance of DNA MTases, no significant differences were observed in methylated sites across the genomes of TSV292 rough and smooth for motifs CACAG, GTWWAC and CATCAG. Despite strain-specific differences in DNA methyltransferase expression, methylation profiles alone may not distinguish co-infecting strains.

## Discussion

Melioidosis causes an estimated 179,000 cases and 90,000 deaths worldwide each year [5]. As *Bp* is intrinsically resistant to antibiotics, treatment relies on last-resort agents. Emerging resistance remains a major concern [9–11]. While simultaneous polyclonal *Bp* infections are rarely reported [26,27], here we report the first such case in Queensland, isolated from a patient swab in 2018.

A key diagnostic challenge revealed by this study is that polyclonality can be mistaken for CMV. Current diagnosis workflows—typically involving (1) single-colony purification with selection of a characteristic wrinkled morphotype on selective media or blood agar; (2) microscopy targeting DNA or RNA sequences using fluorescence in situ hybridization or immunofluorescence; (3) molecular diagnostics such as conventional PCR or sequencing; (4) serodiagnostic tests, including ELISA or agglutination assays; and (5) imaging and spectrometry approaches, including CT or MRI of the patient and MALDI-TOF for bacterial isolates [4]—cannot reliably distinguish *Bp* isolates at the strain level, as these methods generally operate at the genus or species level.

Bacterial polyclonality further complicates these diagnostic approaches. PCR-based assays (qPCR and Illumina sequencing) fail to identify novel variant strains, resulting in amplification bias toward the dominant genotype or a *de novo* hybrid [65]. Misidentification of clonal variants can also stem from mixed DNA templates from polyclonal infections. Serological tests may be influenced by cross-reactive antibodies, as infections with multiple strains can create unclear profiles that hinder interpretation. For instance, if the main strain has a different capsular polysaccharide type, antibody detection might drop below the assay's limit, resulting in false negatives or inconclusive outcomes. Moreover, most diagnostic tests are typically performed on a single pure colony rather than multiple-colonies, making it difficult to distinguish true CMV from mixed-strain infections. This challenge is especially evident when morphological differences are assumed to reflect common CMV, or when a less dominant strain is present that morphologically closely resembles the dominant colony. Overall, this diagnostic blind spot risks underestimating strain diversity and may lead to inappropriate clinical decisions or treatment failures [13].

Because *Burkholderia* species exhibit extensive colony morphotype plasticity with a particular mucoid/wrinkled colony morphotype [14,18,20], relying on phenotypic assessment alone can mask genetically distinct pathogens. In this case, two morphotypes that appeared as rough and smooth variants on CRLA and as mucoid/wrinkled on Ashdown selective medium. The different morphotypes were in fact distinct strains rather than in-host genetic variants that evolved by micro-evolutionary events or products of sample contamination or CMV of a single isolate. Importantly, our experimental procedures were conducted independently from the initial identification of TSV292_1 (rough morphotype) as a novel ST2319 [25]. The smooth isolate, TSV292_2, not only had a distinct ST but also had > 23,000 genomic nucleotide differences, including deletion of multiple genomic regions ranging from ~7–61 kb. While genomic deletions during host adaptation are common, two *Bp* strains isolated ~12 years apart from a single patient with chronic infection were previously shown to differ by only 23 SNPs [55,66]. Therefore, in-host evolution remains unlikely. Cross-contamination of the specimen in the diagnostic laboratory is possible; however, as no other TSV isolates in the strain collection share the same ST as TSV292_2, it is an unlikely but possible event.

We also investigated DNA methylation, given its known role in CMV and intraspecies recombination [67]. While global methylation patterns appeared similar between the isolates, the absence of detectable DNA MTases by LC-MS/MS in the smooth morphotype mirrored our prior findings in smooths CMVs [24]. This could reflect biological differences in MTase stability or abundance but may also stem from limitations of single-replicate methylome analysis in detecting subtle variation.

By integrating genomic, proteomic, and phenotypic analyses, we demonstrate how distinct co-infecting strains can mimic CMV, with smooth and rough colony morphotypes differing in the production of proteins involved in virulence, stress tolerance, drug responses and adaptation—features with direct implications for patient management. A clear example is the trimethoprim-sulphamethadoxone susceptibility test assay. Disk diffusion assays consistently revealed differential susceptibility profiles across all technical and biological replicates, confirming the reliability of our findings under the conditions we used. Treatment could preferentially eliminate the more sensitive isolate and potentially allow proliferation of the resistant strain and lead to treatment failure if not paired with other effective antibiotics. While genomic analysis using the ARDaP tool did not reveal any antibiotic resistance prediction for both isolates, this highlights that genomic data alone may underestimate functional antibiotic resistance.

Our findings also underscore the vulnerability of outbreak tracing and marker gene based phylogenomic inference to hidden co-infection. Whole genome based clonal verification should therefore precede evolutionary or epidemiological analyses of *Bp*. High-quality, closed genomes from long-read assemblies, combined with read-level deconvolution using short-read sequencing, should be standard practice before calling variants. However, if identification of *Bp* is restricted to an analysis of a single colony during diagnosis, polyclonality will be missed.

Covert polyclonality, rather than ultra-rapid evolution, may significantly contribute to the observed genomic diversity. Therefore, enhancing diagnostic procedures by increasing the number of colonies selected and analysed, including those showing CMV, is crucial for *Bp*. Screening multiple colonies has allowed studies in Thailand to successfully identify polyclonal infections with prevalence reaching up to 22% [26,27]. Furthermore, this strain diversity was associated with variation among isolates from related environmental sources [27]. These findings suggest that, in addition to improving diagnostic methods, adopting a One Health approach—characterising the diversity of *Bp* populations in soil and in animals—would greatly improve surveillance efforts and our understanding of infection dynamics [27,68].

Ultimately, this work highlights the urgent need for strain-level diagnostic methods that combine multi-omics and phenotypic data. Developing comprehensive reference databases of these profiles will allow comparative tools to assist in diagnosis, treatment, and surveillance. These advancements will not only enhance patient outcomes but also provide more precise epidemiological insights, deepening our understanding of *Bp* biology.

## Supporting information

**S1 Fig. Genome alignment of TSV292_1 (rough) and TSV292_2 (smooth) with their respective closest isolates genomes.**
(PDF)

**S2 Fig.** MAUVE alignment of both TSV292_1 (rough) and TSV292_2 (smooth) against each other. The red line delimits both chromosomes in each genome. Contig 1 is one the left and contig 2 is on the right. Each color corresponds to an identical region in both genomes.
(PDF)

**S3 Fig.** Genome alignment of TSV292_2 (smooth) on TSV292_1 (rough) using LASTZ alignment tool. Forward alignments between the reference TSV292_1 (rough) contig 1 (A) and contig 2 (B) (x-axis) and TSV292_2 (smooth) query (y-axis) genomes are shown in blue, indicating regions of conserved synteny in the same orientation. Alignments in red represent sequences aligned in the reverse-complement orientation, corresponding to inverted segments relative to the

reference. Gaps with no alignment indicate regions unique to one genome, missing from the other, or below the similarity threshold for alignment.
(PDF)

**S4 Fig.** Genome alignment of TSV292_1 (rough) on TSV292_2 (smooth) using LASTZ alignment tool. Forward alignments between the reference TSV292_2 (smooth) contig 1 (A) and contig 2 (B) (x-axis) and TSV292_1 (rough) query (y-axis) genomes are shown in blue, indicating regions of conserved synteny in the same orientation. Alignments in red represent sequences aligned in the reverse-complement orientation, corresponding to inverted segments relative to the reference. Gaps with no alignment indicate regions unique to one genome, missing from the other, or below the similarity threshold for alignment.
(PDF)

**S5 Fig.** Highlight of the order of genes with an identified as flagellar product. A) TSV292_2 (smooth) genome. B) TSV292_1 (rough) genome.
(PDF)

**S1 Table. Genomic variation and SNPS between TSV292 rough and smooth colonies.**
(XLSX)

**S2 Table. Bp isolates used to generate MLST phylogenic tree.**
(XLSX)

**S3 Table. Genomic region different between TSV292_1 (rough) and TSV292_2 9smooth) isolates.**
(XLSX)

**S4 Table. Predicted genomic islands in TSV292_1 (rough) using Island4 viewer tool.**
(XLSX)

**S5 Table. Predicted genomic islands in TSV292_2 (smooth) using Island4 viewer tool.**
(XLSX)

**S6 Table. Genes predicted as DNA Mtases in TSV292 rough and smooth.**
(XLSX)

**S7 Table. Antibiotic resistance profile of TSV292_1 rough.**
(XLSX)

**S8 Table. Antibiotic resistance profile of TSV292_2 smooth.**
(XLSX)

**S9 Table. Proteomic raw datasearch for TSV282 rough and smooth isolates with homology to Bp K96243.**
(XLSX)

**S10 Table. Proteomic analysis comparing both TSV292_1 rough and TSV292_2 smooth isolates with homology to Bp K96243.**
(XLSX)

**S11 Table. Antibiotic resistance gene prediction.**
(XLSX)

**S12 Table. DNA Methylation prevalence in TSV292 rough and smooth.**
(XLSX)

## Acknowledgments

The authors thank Julien L. Breton--Robin for helping set command lines tools using singularity.

## Author contributions

**Conceptualization:** Pauline M.L. Coulon.

**Formal analysis:** Pauline M.L. Coulon, Piklu Roy Chowdhury.

**Funding acquisition:** Pauline M.L. Coulon, Ian Gassiep, Patrick N.A. Harris, Garry S. A. Myers.

**Investigation:** Pauline M.L. Coulon.

**Methodology:** Pauline M.L. Coulon, Piklu Roy Chowdhury, Kay Ramsay, Aven Lee, Edita Ritmejeryte, Miranda E. Pitt, Joyce To, Sarah Reed.

**Resources:** Sarah Reed, Patrick N.A. Harris, Garry S. A. Myers.

**Visualization:** Pauline M.L. Coulon.

**Writing – original draft:** Pauline M.L. Coulon, Piklu Roy Chowdhury.

**Writing – review & editing:** Pauline M.L. Coulon, Piklu Roy Chowdhury, Ian Gassiep, Kay Ramsay, Aven Lee, Edita Ritmejeryte, Miranda E. Pitt, Joyce To, Sarah Reed, Patrick N.A. Harris, Garry S. A. Myers.

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
