## [Decision Letter · Decision Letter 0]

14 Oct 2025

Dear Dr. Coulon,

Thank you for submitting your manuscript to PLOS ONE. After careful consideration, we feel that it has merit but does not fully meet PLOS ONE’s publication criteria as it currently stands. Therefore, we invite you to submit a revised version of the manuscript that addresses the points raised during the review process.

We look forward to receiving your revised manuscript.

Kind regards,

William C. Nierman, Ph.D.

Academic Editor

PLOS ONE

Journal Requirements:

This work was supported by AIMI and UTS Faculty of Science seed funds to Dr Pauline M.L. Coulon, UTS Strategic Research Accelerator funding (SRA 2726229) to Prof Garry S.A. Myers and EL2 Investigator grant from the NHMRC (APP2033851) to Dr Patrick N.A. Harris. Experimental work was done at UQCCR which would not have been possible without the ISME scholar funding awarded to Dr Pauline M.L. Coulon. P.M.L.C: Concept and visualization, Funding, Experimental work; K.M: Experimental work and reviewing, P.R.C: Genomic comparison; M.P and J.T: DNA libraries and sequencing; A.L, E.R and S.R: sample preparation and LCMS runs for the proteomic experiment; I.G & P.N.A.H: Reviewing and Funding; G.S.A.M: Reviewing and Funding.

This work was supported by AIMI and UTS Faculty of Science seed funds to Dr Pauline M.L. Coulon, UTS Strategic Research Accelerator funding (SRA 2726229) to Prof Garry S.A. Myers and EL2 Investigator grant from the NHMRC (APP2033851) to Dr Patrick N.A. Harris. Experimental work was done at UQCCR which would not have been possible without the ISME scholar funding awarded to Dr Pauline M.L. Coulon.

5. Please ensure that you refer to Figure 4 in your text as, if accepted, production will need this reference to link the reader to the figure.

7. Please remove all personal information, ensure that the data shared are in accordance with participant consent, and re-upload a fully anonymized data set.

Additional guidance on preparing raw data for publication can be found in our Data Policy (https://journals.plos.org/plosone/s/data-availability#loc-human-research-participant-data-and-other-sensitive-data ) and in the following article: http://www.bmj.com/content/340/bmj.c181.long.

Additional Editor Comments :

While the paper strongly supports the conclusion that two distinct genetic isolates are characterized as distinct genetic isolates in the culture collection clinical sample, the author does not address alternative explanations for the presence of the two strains. such as sample contamination, within host divergent evolution, or other possibilities. The author needs to address the concerns and suggestions made by the reviewers.

Reviewers' comments:

Reviewer's Responses to Questions

**Comments to the Author**

1. Is the manuscript technically sound, and do the data support the conclusions?

Reviewer #1: Partly

Reviewer #2: Yes

2. Has the statistical analysis been performed appropriately and rigorously?

Reviewer #1: N/A

Reviewer #2: N/A

3. Have the authors made all data underlying the findings in their manuscript fully available?

Reviewer #1: Yes

Reviewer #2: Yes

4. Is the manuscript presented in an intelligible fashion and written in standard English?

Reviewer #1: Yes

Reviewer #2: Yes

Reviewer #1: The authors report two Burkholderia pseudomallei strains isolated from a single patient at a hospital in Townsville and interpret their findings as evidence of co-infection. This is an interesting and potentially important observation with implications for both the diagnosis and treatment of B. pseudomallei.

However, the evidence as presented does not fully rule out alternative explanations, such as within-host divergent evolution or laboratory contamination. To substantiate the conclusion that these strains represent a true co-infection, further clarification is needed. In particular:

1. From which patient samples were the two strains isolated? Were they derived from the same specimen or from distinct sources?

2. How many independent isolates (colonies) of each strain were recovered?

3. Based on 16S rRNA gene sequences, how distinct are the two strains?

4. From whole-genome sequencing, how extensive are the differences between the strains? Are there genes or long sequence regions unique to one strain, beyond mutations or mobile element–associated insertions/deletions consistent with divergence from a common ancestor? More specifically, what accounts for the larger genome size of TSV292_2 compared with TSV292_1, and does TSV292_1 retain any unique sequences despite its smaller genome?

5. What is the likelihood that one strain (TSV292_1) is closely related to a previously isolated Townsville University Hospital strain? Could this reflect contamination from a preserved isolate rather than both strains representing true infection?

Without addressing these points, the conclusion of co-infection remains plausible but not definitive.

Reviewer #2: Dear Editor,

This manuscript reports the co-isolation of two genetically distinct Burkholderia pseudomallei strains (ST-2319 and ST-2323) from a single melioidosis patient in Townsville, Australia. Using an integrated multi-omics approach (Illumina and Nanopore genome sequencing, proteomics via ZenoTOF 7600 LC-MS/MS, and phenotypic assays), the authors demonstrate that what initially appeared as colony morphotype variation (CMV) actually represented a true polyclonal infection. The work highlights the diagnostic challenges posed by such cases and suggests that standard laboratory workflows may underestimate strain diversity in clinical B. pseudomallei isolates.

This is a technically strong and novel contribution to melioidosis research. It fits PLOS ONE’s scope for methodological rigor and open data sharing.

Here are some comments and suggestions:

Methods clarity:

- Specify sequencing coverage, read depth, and methylation-calling thresholds used for Nanopore data.

- Confirm whether proteomic and phenotypic assays were performed in biological triplicates or single replicates.

- Indicate if any of the reference strains (e.g., B. pseudomallei K96243) was included in phenotypic assays; if not, explain why.

Additional discussions:

- Expand the discussion to address how polyclonality could influence antibiotic therapy outcomes, especially regarding the TMP-SMX resistance observed in the smooth isolate.

- Discuss potential diagnostic challenges for PCR, MALDI-TOF, and serological assays when polyclonal infections occur.

- Soften the statement “covert polyclonality rather than ultrafast evolution is a major driver…” to a more cautious phrasing (e.g., “may contribute significantly to observed genomic diversity”).

- Consider including additional references discussing the prevalence of polyclonal infections, such as, Kaewrakmuk et al PLoS Negl Trop Dis . 2024 Aug 22;18(8):e0012444, when multiple bacterial colonies were selected for genotypings.

**Do you want your identity to be public for this peer review?** For information about this choice, including consent withdrawal, please see our Privacy Policy

Reviewer #1: **Yes: ** Heenam Stanley Kim

Reviewer #2: No

---

## [Author Response · Author response to Decision Letter 1]

16 Nov 2025

Response to Editor and Reviewers’ comments

Firstly, we would like to thank both the Editor and Reviewers for providing valuable feedback, which have improved the manuscript quality. We have addressed and responded to the comments below:

Journal Requirements:

1. Please ensure that your manuscript meets PLOS ONE's style requirements, including those for file naming. The PLOS ONE style templates can be found at https://journals.plos.org/plosone/s/file?id=wjVg/PLOSOne_formatting_sample_main_body.pdfand
https://journals.plos.org/plosone/s/file?id=ba62/PLOSOne_formatting_sample_title_authors_affiliations.pdf

Response: The reviewed manuscript is formatted with the PLOS ONE's style requirements.

Response: Patient consent was waived by the HREC.

Response: Thank you for pointing out the mistake. The correct grant numbers were provided prior to resubmission.

This work was supported by AIMI and UTS Faculty of Science seed funds to Dr Pauline M.L. Coulon, UTS Strategic Research Accelerator funding (SRA 2726229) to Prof Garry S.A. Myers and EL2 Investigator grant from the NHMRC (APP2033851) to Dr Patrick N.A. Harris. Experimental work was done at UQCCR which would not have been possible without the ISME scholar funding awarded to Dr Pauline M.L. Coulon. P.M.L.C: Concept and visualization, Funding, Experimental work; K.M: Experimental work and reviewing, P.R.C: Genomic comparison; M.P and J.T: DNA libraries and sequencing; A.L, E.R and S.R: sample preparation and LCMS runs for the proteomic experiment; I.G & P.N.A.H: Reviewing and Funding; G.S.A.M: Reviewing and Funding.

This work was supported by AIMI and UTS Faculty of Science seed funds to Dr Pauline M.L. Coulon, UTS Strategic Research Accelerator funding (SRA 2726229) to Prof Garry S.A. Myers and EL2 Investigator grant from the NHMRC (APP2033851) to Dr Patrick N.A. Harris. Experimental work was done at UQCCR which would not have been possible without the ISME scholar funding awarded to Dr Pauline M.L. Coulon.

Response: The statement has been reviewed and included into the cover letter as followed: “This work was supported by the funding AIMI and UTS Faculty of Science seed fundings hold by Dr Pauline M.L. Coulon, the strategic research accelerator funding (SRA 2726229) hold by Prof Garry Myers and EL2 Investigator grant from the NHMRC (APP2033851) hold by Dr Patrick N.A. Harris. Royal Australasian College of Physicians Queensland Regional Committee Research Development Grant hold by Ian Gassiep allowed the collection and storage of Bp isolates. Experimental work was done at UQCCR which would not have been possible without the ISME scholar funding awarded to Dr Pauline M.L. Coulon”.

5. Please ensure that you refer to Figure 4 in your text as, if accepted, production will need this reference to link the reader to the figure.

Response: Figure 4 is included in the proteomic result section and can be found at lines 429 and 447.

Response: Captions of supporting information are provided at the end of the manuscript for both supplementary figures and tables.

7. Please remove all personal information, ensure that the data shared are in accordance with participant consent, and re-upload a fully anonymized data set.

Response: Our data do not contain any personal information.

Response: As suggested by Reviewer 2, a relevant missing article has been cited and included in the revised manuscript.

Additional Editor Comments :

While the paper strongly supports the conclusion that two distinct genetic isolates are characterized as distinct genetic isolates in the culture collection clinical sample, the author does not address alternative explanations for the presence of the two strains. such as sample contamination, within host divergent evolution, or other possibilities. The author needs to address the concerns and suggestions made by the reviewers.

Response: Thank you for your valuable comment, which aligns with Reviewer 1’s feedback. We have included alternative explanations for the presence of the two strains in the Discussion section, starting at line 510.

Reviewers' comments:

Reviewer #1:

The authors report two Burkholderia pseudomallei strains isolated from a single patient at a hospital in Townsville and interpret their findings as evidence of co-infection. This is an interesting and potentially important observation with implications for both the diagnosis and treatment of B. pseudomallei.

However, the evidence as presented does not fully rule out alternative explanations, such as within-host divergent evolution or laboratory contamination. To substantiate the conclusion that these strains represent a true co-infection, further clarification is needed. In particular:

1. From which patient samples were the two strains isolated? Were they derived from the same specimen or from distinct sources?

Response: Thank you for your questions. Our study is based on the isolation of two different morphotypes from the same glycerol stock, which was obtained after selection and identification of B. pseudomallei from a blood culture of an infected patient. The culture was used to set up a purity plate, and the resulting growth was used to make the glycerol stock. As the requested information is indeed relevant, we have now included these details in the Methods section (line 102).

2. How many independent isolates (colonies) of each strain were recovered?

Response: We isolated four colonies of each morphotype (rough and smooth) from streaked plates prepared directly from the −80 °C stock. This information has been added to the Methods section (line 102). While we did not quantify the proportion of each morphotype, visual inspection of the streaked plates (Fig. 1A) suggests that TSV292_1 rough colonies are more prevalent than TSV292_2 smooth ones.

3. Based on 16S rRNA gene sequences, how distinct are the two strains?

Response: Both isolates were initially identified using MLST sequencing, which offers higher resolution for distinguishing strains within the same species. In response to this comment, we also examined the 16S rRNA sequences and found that three of the four copies carried a G→A mutation at position 182 bp, indicating minor variation at the 16S level. Alignment of the MLST sequences revealed SNPs within several genes, resulting in the identification of distinct sequence types (STs).

4. From whole-genome sequencing, how extensive are the differences between the strains? Are there genes or long sequence regions unique to one strain, beyond mutations or mobile element–associated insertions/deletions consistent with divergence from a common ancestor? More specifically, what accounts for the larger genome size of TSV292_2 compared with TSV292_1, and does TSV292_1 retain any unique sequences despite its smaller genome?

Response: Thank you for highlighting these questions. To address them, we have added the following analysis to the manuscript. We cross-compared the annotated gene sets from both assemblies by aligning each against the nucleotide sequences of the opposite isolate using BLASTn. Results with >90% sequence identity were retained and further cross-analyzed using the equivalent approach against the B. pseudomallei K96243 genome. Genes located within mobile element–associated insertion/deletion regions were subsequently excluded.

This analysis identified only one unique gene in TSV292_R (TSV292_R_RS_00840), predicted to encode a hypothetical protein. According to the literature, this gene is downregulated when B. pseudomallei SZ5028 is grown in minimal medium with arabinose compared to glucose (DOI: 10.1128/IAI.72.7.4172–4187.2004).

Furthermore, the difference in genome size between TSV292_1 and TSV292_2 is likely explained by the total size of the deleted/inserted genomic regions (94 kb in TSV292_1 and 357,022 bp in TSV292_2), as well as the number of duplicated genes (240,390 bp in TSV292_1 and 74 kb in TSV292_2). The methodology and corresponding results have now been detailed in the manuscript (lines 162, 352, and 376).

5. What is the likelihood that one strain (TSV292_1) is closely related to a previously isolated Townsville University Hospital strain? Could this reflect contamination from a preserved isolate rather than both strains representing true infection?

Response: Thank you for your feedback regarding the potential for contamination. Isolate TSV292 was initially cultured, and a rough colony was sequenced, which resulted in a new sequence type (ST) described by Gasseip et al. (2024) [25]. The same study identified this ST within a cohort of approximately 200 TSV isolates.

For the present study, all experimental steps—from streaking the isolate stock to genomic DNA extraction—were performed by a different researcher, and sequencing was conducted independently at another facility using both Illumina and Nanopore technologies. The resulting ST for TSV292_1 matched the previously described ST2323 [25], whereas TSV292_2 represented a novel ST, differing by over 22,000 mutations across the genome – alongside with deletions of multiple genomic regions between both isolates.

Illumina reads were cross-mapped onto the assembled genome, and clear single-nucleotide polymorphisms (SNPs) were identified in MLST genes, each supported by at least 90% (often 100%) of the reads. These findings strongly suggest a genuine polyclonal infection rather than contamination. Nevertheless, we have added to the Discussion a consideration of alternative explanations that could account for the presence of two distinct isolates (from line 510).

Without addressing these points, the conclusion of co-infection remains plausible but not definitive.

Reviewer #2:

Dear Editor,

This manuscript reports the co-isolation of two genetically distinct Burkholderia pseudomallei strains (ST-2319 and ST-2323) from a single melioidosis patient in Townsville, Australia. Using an integrated multi-omics approach (Illumina and Nanopore genome sequencing, proteomics via ZenoTOF 7600 LC-MS/MS, and phenotypic assays), the authors demonstrate that what initially appeared as colony morphotype variation (CMV) actually represented a true polyclonal infection. The work highlights the diagnostic challenges posed by such cases and suggests that standard laboratory workflows may underestimate strain diversity in clinical B. pseudomallei isolates.

This is a technically strong and novel contribution to melioidosis research. It fits PLOS ONE’s scope for methodological rigor and open data sharing.

Here are some comments and suggestions:

Methods clarity:

- Specify sequencing coverage, read depth, and methylation-calling thresholds used for Nanopore data.

Response: Thank you for noting the missing information regarding our sequencing data and methylation analysis parameters. Details on sequencing coverage and read depth have been provided in the “Genome assembly” method section (line 126), and the methylation-calling thresholds and related parameters are described in the “Methylome” method section (line 175).

- Confirm whether proteomic and phenotypic assays were performed in biological triplicates or single replicates.

Response: The number of replicates for phenotypic assays (4; line 253) and proteomic analyses (3) were originally included in the method. However, we have added a subsection to described how the samples were cultured for proteomic analyses (line 183)

- Indicate if any of the reference strains (e.g., B. pseudomallei K96243) was included in phenotypic assays; if not, explain why.

Response: Thank you for your comment. The reference isolate Bp K96243 was not included in our tests as, phenotypes are strain-specific. The same statement was added lines 260 and 276.

Additional discussions:

- Expand the discussion to address how polyclonality could influence antibiotic therapy outcomes, especially regarding the TMP-SMX resistance observed in the smooth isolate.

Response: We have now included discussion relating to antibiotic resistance profile differing between both isolates and the potential consequences (line 533).

- Discuss potential diagnostic challenges for PCR, MALDI-TOF, and serological assays when polyclonal infections occur.

Response: Thank you for the suggestion, this has been added to the discussion from line 493.

- Soften the statement “covert polyclonality rather than ultrafast evolution is a major driver…” to a more cautious phrasing (e.g., “may contribute significantly to observed genomic diversity”).

Response: The statement has been changed as requested and can be found line 544.

- Consider including additional references discussing the prevalence of polyclonal infections, such as, Kaewrakmuk et al PLoS Negl Trop Dis . 2024 Aug 22;18(8):e0012444, when multiple bacterial colonies were selected for genotypings.

Response: Thank you for providing the reference. We have included it into the paper, in the abstract, introduction and discussion under ref [27] (lines 24; 82; 554).

---

## [Editor Report · Decision Letter 1]

20 Nov 2025

Co-isolation of genetically distinct Burkholderia pseudomallei strains from a single patient in North Queensland

PONE-D-25-51165R1

Dear Dr. Coulon,

We’re pleased to inform you that your manuscript has been judged scientifically suitable for publication and will be formally accepted for publication once it meets all outstanding technical requirements.

Kind regards,

William C. Nierman, Ph.D.

Academic Editor

PLOS ONE
---

## [Editor Report · Acceptance letter]

PONE-D-25-51165R1

PLOS One

Dear Dr. Coulon,

I'm pleased to inform you that your manuscript has been deemed suitable for publication in PLOS One. Congratulations! Your manuscript is now being handed over to our production team.

Kind regards,

on behalf of

Dr. William C. Nierman

Academic Editor

PLOS One